# The Antioxidative Effects of Flavones in Hypertensive Disease

**DOI:** 10.3390/biomedicines11112877

**Published:** 2023-10-24

**Authors:** Alexandria Porcia Haynes, Selam Desta, Taseer Ahmad, Kit Neikirk, Antentor Hinton, Nathaniel Bloodworth, Annet Kirabo

**Affiliations:** 1Division of Clinical Pharmacology, Department of Medicine, Vanderbilt University Medical Center, 2215 Garland Avenue, P415C Medical Research Building IV, Nashville, TN 37212, USA; alexandria.p.haynes@vumc.org (A.P.H.); selam.desta@bison.howard.edu (S.D.); taseer.ahmad@vumc.org (T.A.); 2Department of Biology, College of Arts and Sciences, Howard University, Washington, DC 20059, USA; 3Department of Pharmacology, College of Pharmacy, University of Sargodha, University Road, Sargodha 40100, Punjab, Pakistan; 4Department of Molecular Physiology and Biophysics, Vanderbilt University, 2201 West End Ave, Nashville, TN 37235, USA; kit.neikirk@vanderbilt.edu (K.N.); antentor.o.hinton.jr@vanderbilt.edu (A.H.)

**Keywords:** diosmetin, flavone, flavonoid, hypertension, inflammation, NADPH oxidase, protein kinase C (PKC), molecular pharmacology

## Abstract

Hypertension is the leading remediable risk factor for cardiovascular morbidity and mortality in the United States. Excess dietary salt consumption, which is a catalyst of hypertension, initiates an inflammatory cascade via activation of antigen-presenting cells (APCs). This pro-inflammatory response is driven primarily by sodium ions (Na^+^) transporting into APCs by the epithelial sodium channel (ENaC) and subsequent NADPH oxidase activation, leading to high levels of oxidative stress. Oxidative stress, a well-known catalyst for hypertension-related illness development, disturbs redox homeostasis, which ultimately promotes lipid peroxidation, isolevuglandin production and an inflammatory response. Natural medicinal compounds derived from organic materials that are characterized by their anti-inflammatory, anti-oxidative, and anti-mutagenic properties have recently gained traction amongst the pharmacology community due to their therapeutic effects. Flavonoids, a natural phenolic compound, have these therapeutic benefits and can potentially serve as anti-hypertensives. Flavones are a type of flavonoid that have increased anti-inflammatory effects that may allow them to act as therapeutic agents for hypertension, including diosmetin, which is able to induce significant arterial vasodilation in several different animal models. This review will focus on the activity of flavones to illuminate potential preventative and potential therapeutic mechanisms against hypertension.

## 1. Introduction

Hypertension is the single most important cause of cardiovascular disease and premature death in the world, affecting 1.4 billion people and incurring an estimated annual cost of $46 billion [1]. Nearly half of the adults in the United States (47%) develop hypertension, defined as a systolic blood pressure of 130 mmHg or higher [2,3]. A major risk factor for hypertension is excess dietary salt intake. The American Heart Association (AHA) recommends for individuals to consume a maximum of 2300 mg of salt per day; however, less than 10% of the United States population observes this recommendation [4]. The mechanisms describing the pathogenesis of hypertension are variable, but emerging data in mice and humans implicate immune cell activation as a key contributor to salt-sensitivity in blood pressure.

The initiation and maintenance of hypertension are dependent on inflammation caused by activated immune cells [5]. Our lab has previously found that salt induces immune cell activation via the induction of reactive oxygen species (ROS) and lipid peroxides, a mechanism summarized in Figure 1. Sodium ions (Na^+^) enter antigen-presenting cells (APCs) through the epithelial sodium channel (ENaC) and incite a further influx of calcium via the Na⁺/Ca^2^⁺ ion exchanger. Elevated intracellular Ca^2+^ levels activate PKC to induce phosphorylation of p47phox, a critical component of the NADPH oxidase complex responsible for its assembly and activation [6]. NADPH oxidase generates ROS, which in turn activates the NLRP3 inflammasome and stimulates the subsequent production of pro-inflammatory cytokines [7]. Elevated NADPH oxidase activity also induces the formation of a class of lipid peroxides through ROS-induced peroxidation of arachidonic acid. These lipid peroxides, called Isolevuglandins (IsoLGs), are nucleophiles that preferentially react with lysine residues on proteins to form lipid-protein adducts. These adducts can be proteolyzed and presented to patrolling T-cells [6,7]. These important mechanisms, while critical for hypertension pathogenesis, are currently unaddressed by conventional therapies; thus, they represent a unique opportunity for impactful pharmacologic interventions.

Flavonoids and flavonoid-derived compounds represent one such potential intervention. Flavonoids are natural phenolic compounds found in fruits, vegetables, wines, flowers, tea, and other natural products [8]. They are known for their anti-inflammatory, anti-oxidative, and anti-mutagenic effects. The structure of flavonoids consists of a phenyl ring, B, a benzene ring, A, and a heterocyclic C ring (Figure 2). Flavonoids are divided into different subgroups based on their structure, including flavonols, flavones, flavanones, flavanols, flavanonols, and isoflavones (Figure 3) [8]. Flavonols and flavones play a crucial role against ROS as a source of external antioxidants due to their anti-radical capabilities, which occur in a structurally dependent manner, with C4 and C3 hydroxyl groups associated with stronger scavenging potential [9]. Flavones are the most hydrophobic of the flavonoid subgroups, one of the qualities that indicate the membrane permeability of the flavonoid, which is important to the exertion of therapeutic effects [10]. Many flavonoids are thought to lower blood pressure and aid in hypertension. Diosmetin (5,7-dihydroxy-2-(3-hydroxy-4-methoxyphenyl)chromen-4-one), a flavone, induces significant arterial vasodilation in hypertensive rats [11]. In this review, we discuss the pharmacologic properties of flavones and their antioxidative effects on pro-inflammatory pathways implicated in hypertension.

## 2. Flavonoids Exert Their Effects by Targeting Intracellular Proteins

Although flavonols are more active antioxidants than flavones, the metabolic stability and membrane permeability are much higher in flavones [9,12]. This and the ability to produce a physiologic response depends primarily on its hydrophobicity [13]. A partition coefficient (PC) test using an n-octanol/HEPES solution determined the hydrophobicity of four flavonoids, eriodyctiol, quercetin, luteolin, and taxifolin. Of the four flavonoids tested, luteolin had the highest affinity for octanol, making it the most hydrophobic, followed by quercetin, eriodyctiol, and then taxifolin [14]. The hydrophobicity of flavonoids is determined by the number of hydroxyl functional groups attached to the rings, as well as the number and placement of pi bonds in the rings. Flavonols, like quercetin, and flavanonols, like taxifolin, have similar structures. Of these two types of flavonoids, flavanonols have fewer pi bonds, making them less hydrophobic. Flavones, like luteolin, have fewer hydroxyl groups, causing them to be more hydrophobic. Methoxy flavones, like diosmetin, are considered the most hydrophobic of the flavonoids, with correspondingly higher metabolic stability and membrane transport ability when compared with other flavonoids (Figure 4) [12]. Although flavonoid hydrophobicity correlates with its ability to pass through the cell membrane, whether this correlates with physiological activity has yet to be fully discovered. Diosmetin has vasorelaxation effects that are over 10 times greater than verapamil, a potent vasodilatory agent used to treat hypertension, due to its greater inhibitory effect on Ca^2^⁺ release from intracellular Ca^2^⁺ stores [11].

## 3. Flavones Target NADPH Oxidase Transcription by Inhibiting Smad3 Phosphorylation

NADPH oxidase expression is required for the initiation and maintenance of hypertension, making the pathways regulating NADPH oxidase expression important therapeutic targets. Smad3 is an important transcriptional regulator of NADPH oxidase gene expression in APCs. Its phosphorylation and subsequent activation are regulated by a variety of pathways upstream [15]. IL-6 signaling, via the JAK2/STAT3 pathway, results in Smad3 phosphorylation, which leads to the upregulation of NADPH oxidase components [15,16]. The canonical TGF-β1 signaling pathway, which is also involved in the activation of the pro-inflammatory transcription factor NF-κB, is dependent on Smad3 phosphorylation [17]. A study done by Li et al. used an MTT assay and western blot to reveal the effect of apigenin on TGF-β1 and Smad3 protein levels, respectively [18]. Apigenin, a flavone, prevented TGF-β1-induced Smad3 phosphorylation, possibly by inhibiting JAK2/STAT3 signaling [18]. A different study conducted by Ning et al. used western blot to reveal that the application of diosmetin in µM amounts decreased the expression of phosphorylated STAT3 in human osteosarcoma cells [19].

Pectolinarigenin treatment, another dimethoxy flavone, is implicated in the suppression of the TGF-β/Smad3 and the JAK2/STAT3 pathways in mice with hyperuricemic nephropathy [20]. In a study by Ren et al., western blot analysis showed that pectolinarigenin treatment decreased TGF-β expression and Smad3 phosphorylation. Pectolinarigenin treatment was also shown by this study to significantly decrease IL-6 protein levels and STAT3 phosphorylation, which would decrease JAK2/STAT3 signaling [20]. Together, this indicates that certain flavones can alter NADPH oxidase activity, and the associated inhibition of hypertension, through Smad3 inhibition.

Flavones potentially interact with other genes or proteins involved in TGF-β1 activation to inhibit Smad3 phosphorylation. A study carried out by Zhang et al. hypothesized that RGMa may play an integral role in TGF-β1-mediated phosphorylation of Smad3 [21] This study used western blot and immunofluorescence staining to find that as the expression of TGF-β1 increases, its surface receptor is activated, which enhances RGMa expression. RGMa forms a complex with Smad3 and the TGF-β1 receptor, facilitating Smad3 phosphorylation. This study further showed that RGMa inhibition was correlated with a reduction in Smad3 phosphorylation [21]. A study by Arango et al. utilized second-generation (PD) sequencing, a method that discovers small molecule–protein interactions, to identify RGMa as a target gene for apigenin [22]. In another study, Gao et al. found that apigenin improves hypertension in spontaneously hypertensive rats by down-regulating NADPH oxidase-dependent ROS expression [23]. These findings suggest that apigenin inhibits NADPH oxidase-mediated ROS production, potentially by interacting with RGMa to regulate Smad3 phosphorylation, showing how cardiac hypertrophy may be modulated on the basis of apigenin-dependent inflammation and ROS.

Resveratrol is a polyphenol found in grapes and peanuts that is known for its biological activities and pharmacological effects, including inhibition of TGF-β1-mediated epithelial-mesenchymal transition (EMT) in breast cancer, which is mediated by Smad3 phosphorylation [24,25]. Since the structure of resveratrol is similar to the structure of flavones, this inhibition suggests the importance of the hydroxyl group on the 4′ carbon of the phenyl ring, as well as the 5 carbon and 7 carbon of the benzene ring because both compounds share these groups.

Diosmetin can also inhibit protein lysine-specific histone demethylase 1A (LSD1A), expressed by KDM1A, which in turn regulates TGF-β1 production [26]. LSD1A is an epigenetic regulator that promotes EMT in various kinds of cancer [27]. EMT allows solid cancer cells to migrate by repressing epithelial markers and activating mesenchymal markers. LSD1A aids in this process by facilitating heterochromatin demethylation and euchromatin methylation, which activates the transcription of oncogenes and suppresses the expression of tumor suppressor genes, respectively [28], thereby leading to TGF-β-mediated EMT [29]. This shows that LSD1A is involved in the activation of TGF-β-mediated EMT, which is dependent on Smad3 phosphorylation [30]. LSD1A is a target for diosmetin, which potentially elucidates the inhibitory effects of flavones on TGF-β1-mediated Smad3 phosphorylation (Figure 5).

## 4. Flavones Affect Mitochondrial Biogenesis, Dynamics, and Energetics

Given the roles of flavones in NADPH oxidase expression, there are also potential roles in the mitochondrial formation of signaling factors. Mitochondria are known to play a role in hypertension, with their dysfunction being linked to inflammation in cardiac tissue [31]. Flavones can alter mitochondrial metabolism through action sites that are located between complexes I and III, conducive to changes in mitochondrial membrane properties [32]. Novel anti-tumor drugs have incorporated flavone side chains in the structure, which can selectively generate ROS in hepatoma cells through preferential targeting of mitochondria [33]. In other conditions, flavones can protect against oxidative stress and inhibit myocardial ischemia, suggesting that flavone derivatives mechanistically target mitochondria with disease-dependent responses [34]. This underscores the importance of further understanding the pluralistic roles of flavones in ROS generation, which may be dependent on how flavones are implicated in the mitochondrial structure or the type of flavone.

Salvigenin, a trimethoxylated flavone, not only stimulated mitochondria but also decreased lipid levels to protect against metabolic syndrome [35]. Given the antimetastatic nature of flavonoid compounds, through induction of mitochondrial-mediated apoptosis [33], the roles of flavones in mitochondria-mediated hypertension need to be better elucidated. Flavones have been implicated in the metabolic process through mitochondrial alteration. Specifically, sudachitin, fruit-derived flavone, reduced weight gain in high-fat diet mice through mitochondrial biogenesis, resulting in improved insulin resistance [36]. In skeletal muscle, it is possible that flavones interact with Sirt1 and PGC-1α to mediate mitochondrial dynamics to protect against metabolic disorders [36]. Similarly, epigallocatechin-3-gallate, a green-tea-derived flavone, increases mitochondrial biogenesis through the activation of AMPK [37]. As previously reviewed, AMPK can both upregulate PGC-1α as well as NADH-mediated Sirt1 [38]. This suggests a pathway in which flavones can modulate mitochondrial biogenesis through the upregulation of PGC-1α, leading to NRF1 and NRF2, and subsequently TFAM activation to result in mtDNA-mediated mitochondrial biogenesis [38]. However, as previously reviewed, flavones are unique from other categories of flavonoids, including flavonols, flavanones, isoflavone, anthocyanins, and chalcones, which may play unique roles in mitophagy [38]. One key aspect in which flavones may uniquely modulate mitochondrial-dependent hypertension is through mitochondria fusion.

It is important to consider how flavones may impact mitochondria through pro-fission or pro-fusion factors, which can alter the relative mitochondrial dynamics that govern mitochondria biogenesis and subsequent energetics. Xanthohumol, another flavone, has been reported to upregulate mitofusin 2, a mitochondrial profusion factor, to alleviate murine neuronal death in nervous system diseases [39]. Similarly, while hypertension can be spawned by reduced mitochondria fusion, flavonoids have been shown to reverse Angiotensin II-dependent increases in dynamin-related protein 1 (profission) and decreases in optic atrophy 1 (profusion) [40]. These mitochondrial structural changes may have functional impacts on flavone-mediated hypertension.

Specifically, diosmetin treatment in colitis shows interaction with Sirt1 along with decreased inflammation [41], suggesting mitochondria biogenesis-linked resistance to hypertension. Similarly, in a cancer model, through the upregulation of p53 and similar pathways, Diosmetin results in potential mitochondrial membrane alteration to cause apoptosis [42]. Similarly, Diosmetin also has cytoprotective effects in myocardial ischemia injury through decreasing oxidative stress [43]. The modulation of oxidative stress and membrane potential by diosmetin suggests that it also causes structural changes that may impact bioenergetic-dependent hypertension development; however, it is unclear if diosmetin specifically modulates mitochondria biogenesis and structure.

## 5. Flavones Inhibit PI3Kγ, PKC, and Intracellular Ca^2^⁺ Release

When excess Ca^2+^ enters the cell via the Na⁺/Ca^2^⁺ ion exchanger, the increase in Ca^2+^ concentration activates PKC, which phosphorylates p47phox to activate NADPH oxidase [6]. This makes PKC a necessary enzyme for the activation of NADPH oxidase. Flavonoids interact with a number of protein kinase signaling cascades, including the PKC pathway [44]. Flavones and flavonols are the most potent flavonoid PKC inhibitors of PKC due to the double bond and hydroxyl present in the flavonoid heterocyclic ring. However, since diosmetin has a methoxy group on the carbon 4′ of the phenyl ring, this could potentially affect its ability to inhibit PKC [45]. Despite this limitation, diosmetin is more hydrophobic and could better permeate cell membranes than other flavones with a hydroxyl group on the 4 carbons of the phenol ring, such as luteolin or apigenin (Figure 2). Diosmetin prevents vasoconstriction by inhibiting the release of Ca^2^⁺ from intracellular Ca^2^⁺ stores, which increases intracellular Ca^2^⁺, thereby activating PKC [11].

PI3Kγ activation in immune cells is known to assemble and activate NADPH oxidase, leading to ROS production [46,47]. Genkwanin, a methoxy flavone found in several plant species, inhibits the expression of PI3Kγ and interacts in its ATP binding pocket [48]. Another methoxy flavone known as acacetin was shown to bind in the ATP binding pocket of PI3Kγ using hydrophobic interactions with Lys-833 and Asp-964, important residues for catalytic activity [49]. This binding activity is similar to that of known inhibitors of PI3Kγ, and causes a significant decrease in activity [49].

A study conducted by Ahmad et al. tested juglone, an organic compound originating from a walnut tree, for its vasodilatory and potential anti-hypertensive properties. The study used injections into normotensive and hypertensive rats to establish that juglone also acts as a vasorelaxant, partly due to its ability to inhibit the release of Ca^2^⁺ [50]. Since juglone can act as a vasorelaxant similar to diosmetin, there is suggested importance of their shared functional groups. Both molecules have a carbonyl group and a hydroxyl group in similar places, indicating that these groups and positions are necessary for their vasorelaxant properties [45]. Another study by Ahmad et al. highlighted diosmetin’s ability to inhibit the vasocontraction caused by the activation of PKC, which suggests that diosmetin inhibits PKC (Figure 6) [51]. However, the interplay between PKC inhibition and PI3Kγ inhibition remains unclear because it may be possible that both factors are inhibited to reduce NADPH oxidase activity, or if these serve as two alternative antioxidant pathways.

## 6. Flavones and MERCs

Given that flavones affect intracellular Ca^2^⁺ release, mitochondria endoplasmic reticulum contact sites (MERCs) may have unique functional and structural impacts by flavones. MERCs are organelle–organelle contacts that facilitate numerous functions including calcium homeostasis and phospholipid remodeling [52]. MERCs can be modulated through the PI3K/AKT/mTOR pathway, with specifically mammalian TOR complex 2 controlling MAM integrity [53]. Notably, Dracocephalum moldavica L flavones can protect ischemic mitochondria through PI3K/AKT/mTOR pathways [34]. Alternatively, flavonoids such as Nobiletin have been seen to modulate ER stress through the downregulation of PI3K/AKT/mTOR pathways to cause MERC-induced mitochondria dysfunction in cancers [54]. Similarly, in cancers, flavones have been shown, through polyhydroxylation at positions 3 and 6, to inhibit sarcoplasmic reticulum Ca^2+^-ATPases, thus inhibiting mitochondrial calcium transfer [55]. Furthermore, kaempferol inhibits human osteosarcoma through increases in Ca^2+^ with reduced mitochondrial membrane potentials [56]. For WJ9708012, a methoxyflavanone derivative, this anti-cancer pathway has been elucidated as ER-stress dependent increases in GADD153 and GRP78, concomitant with PKC-α-mediated mitochondria stress, resulting in mTOR pathway alterations to cause apoptosis [57]. These pluralistic roles of interacting with the mTOR pathway underscore how MERCs may have differential roles that vary on the basis of flavone type.

Conversely, in healthy states, flavones can aid in the formation of MERCs for healthy calcium transfer. While research is limited, Luteolin, a flavone, has been observed to increase mitochondrial function independent of biogenesis concomitantly with MERC-dependent regulation of intracellular Ca^2+^ content [58]. Thus, given the demonstrated role of flavones in MERC formation, it is possible that Ca^2+^ levels are modulated by MERCs to help determine hypertension. Notably, in pancreatic cancers, another flavone, fisetin, may induce p8-dependent autophagy with interactions in AMPK/mTOR pathways [59]. Another study found that fisetin enhanced cardiac function to protect against hypertension [60], suggesting MERCs as an underexplored target in flavone-mediated hypertension.

Notably, in the context of Diosmetin, past studies have shown that its treatment may increase activation of PI3K-Akt [61], suggesting that it may modulate calcium homeostasis through mTOR-dependent MERC formation. However, whether specific compounds on flavones promote MERC tethering remains unclear, which could be strongly elucidated by the usage of proximity ligation assay [62] or 3D reconstruction [63] to assess MERC distances following flavone treatment.

## 7. Flavones Scavenge ROS through Activating the Nrf2 Transcription Factor

NADPH oxidase-generated ROS is another potential therapeutic target for hypertension. Nrf2 is a transcription factor that controls the expression of antioxidant genes, which are essential for intracellular redox homeostasis and inflammation regulation [64]. Nrf2 achieves this by enhancing the expression and function of heme oxygenase 1 (HO-1), an antioxidant capable of degrading heme and scavenging ROS [65]. Scavenging of ROS prevents the activation of the NLRP3 inflammasome, reducing the inflammation response associated with hypertension [7]. The activation of Nrf2 neutralizes ROS associated with NADPH oxidase activation. Liu et al. found that diosmetin greatly increased activity of the Nrf2/HO-1 pathway in the presence of lipopolysaccharide (LPS)-induced oxidative stress. Treatment with diosmetin also decreased NLRP3 inflammasome activation, suggesting a decrease in cytokine production [7,66]. A different study suggested that quercetin treatment yielded comparable results, where the intracellular anti-oxidative activity and ROS scavenging ability of flavonoids was revealed to be completely due to Nrf2 activation rather than the natural anti-oxidative capacity of flavonoids [65]. However, diosmetin is better able to penetrate the cell than quercetin because it is more hydrophobic; some of the inhibitory effects of diosmetin may be due to its antioxidant abilities (Figure 7). This effect of methoxy flavone physiological antioxidant activity through increased activity of the Nrf2/HO-1 pathway, as well as the direct radical scavenging ability, is seen in pectolinarigenin. A study done by Shiraiwa showed that pectolinarigenin treatment increased protein expression of Nrf2 and HO-1 and had direct ROS scavenging capabilities when administered to mice orally [67].

Diosmetin exhibits inhibitory effects on the NLRP3 inflammasome, along with having blood pressure lowering effects similar to that of captopril, a hypertension treatment [68]. The flavone genkwanin has also been shown to inhibit NLRP3 inflammasome protein expression [48]. MPP+ induces activation of the NLRP3 inflammasome through increased expression of NADPH oxidase [69]. In western blot performed by Li et al., genkwanin was shown to inhibit MPP+ induced activation of the NLRP3 inflammasome [70].

## 8. Flavones Act as an AhR Agonist and Inhibits CYP1A1 Activity

Along with NADPH oxidase, the aryl hydrocarbon receptor (AhR) represents another avenue for pharmacological modulation of inflammation in hypertension [71]. AhR is closely related to inflammation, oxidative stress, and blood pressure regulation. A study by Lund et al. measured the blood pressure of AhR−/− and wild-type (WT) mice at 225 m and 1632 m above sea level after 11 days. The change in blood pressure for the AhR−/− mice was significantly greater than the change in blood pressure for the WT mice [72]. Furthermore, Coelho et al. found that chronic intermittent hypoxia (CIH)-induced hypertension in mice resulted in an overexpression of CYP1A1, which is an indicator of AhR activation, in the kidney along with an increase in blood pressure after 21 days. An AhR inhibitor stopped this increase in blood pressure [73]. Flavonoids, specifically pentahydroxy, hexahydroxy, and tetra/trihydroxy flavonoids, are able to bind to AhR as agonists or antagonists depending on their structure and orientation in the binding pocket. Pentahydroxy flavonoids, such as quercetin, tend to bind as an AhR agonist and orient themselves to associate with amino acid residues known to activate CYP1A1 induction. Tetra/trihydroxy flavonoids, such as apigenin and diosmetin, tend to bind as AhR antagonists and orient themselves to associate with amino acid residues that do not activate CYP1A1, 2–4-fold [74]. Since tetra/trihydroxy flavonoids act as antagonists, they are potentially useful in stopping hypertension associated with chronic hypoxia. Diosmetin targets AhR and its downstream product CYP1A1. AhR responds to carcinogens by releasing CYP1A1, which metabolizes them. Carcinogen breakdown leads to a buildup of ROS. Upregulation of CYP1A1 due to carcinogen exposure causes the unregulated release of ROS disrupting the redox balance in the cell, which leads to the oxidation of arachidonic acid and ultimately hypertension [75] Diosmetin is a potent inhibitor of CYP1A1, an interaction that would inhibit the metabolism of hydrocarbons, the production of excess ROS, and the oxidation of arachidonic acid that leads to hypertension (Figure 8) [76]. Luteolin, a flavone and tetra/trihydroxy flavonoid, has also been shown by Zhang et al. to be a potent AhR antagonist [77]. Flavones act uniquely as AhR antagonists to stop blood pressure increases compared to other flavonoids and inhibit CYP1A1.

## 9. Flavone Inhibition of MRP-1

Diosmetin is also responsible for inhibitory effects on angiotensin II-induced hypertension through the inhibition of multidrug resistance protein 1 (MRP-1). MRP-1 plays a significant role in modulating intracellular glutathione (GSH) levels. In cells, GSH scavenges lipid peroxides, such as IsoLGs, which protect against ROS. Once GSH scavenges these lipid peroxides, it is oxidized to GSSG and must be recycled using GSSG reductase. Before this process can happen, MRP-1 shuttles GSSG out of the cell so it cannot be recycled, which increases ROS presence, NADPH oxidase subunit expression, and ultimately blood pressure. Widder et al. showed that knockout of the MRP-1 gene inhibited hypertension [78]. Diosmetin can bind to and inhibit MRP-1, which potentially explains its vasodilating effects [26]. Because MRP-1 is activated by ATP, diosmetin potentially binds where ATP should bind and acts as an antagonist, which is the same way that diosmetin acts as an antagonist for the protein IPMK [79]. The diosmetin inhibition of MRP-1 allows GSH to scavenge ROS and prevent oxidative stress associated with hypertension (Figure 9).

## 10. Conclusions

The vasodilating and calcium signaling antagonist effect may justify the use of flavones as a future therapeutic agent for hypertension. Flavones, including diosmetin and apigenin, have multiple targets within the hypertensive pathway that potentially explain this effect (Figure 10). A lot of these targets lead to decreased production or activity of ROS from NADPH oxidase. However, NADPH oxidase can be a problematic target because targeting this molecule is associated with increased risks of infection and autoimmune disorders [80]. More information is needed about the intricate pathways of NADPH oxidase in hypertension to address these risks, but because a lot of flavones are promiscuous inhibitors and bind where ATP would, this could complicate the use of flavones as therapeutic agents [81]. How and where binding of flavones occurs, as well as the necessary functional groups, remain elusive but would aid in the use of these natural products against hypertension.

The vasodilating and antioxidant effects of flavones could also be explained by their ability to induce antioxidant enzymes such as superoxide dismutase, which breaks down ROS, glutathione peroxidase, which increases GSH levels, and HO-1, which scavenges ROS [82]. Although this has been shown to be ultimately beneficial, overproduction of HO-1 and GSH are also associated with various types of cancer and chemotherapy resistance [83]. Further research is necessary to determine if flavones play a role in chemotherapy resistance, but the use of diosmetin as a therapeutic agent against oxidative stress has been shown to not affect endothelial cellular metabolism or the stability of endothelial cell membranes at concentrations up to 250 µM [82].

Although the pathways require future experiments for confirmation, flavones undoubtedly mitigate hypertension and hypertension pathogenesis. They do this due to their hydrophobicity, ability to permeate the cell, and ability to modify cellular protein activity. Although diosmetin and other flavones can target MRP-1, inhibit intracellular Ca^2+^ increase, and reduce NADPH oxidase-mediated ROS production, further research is needed to connect them to intracellular GSH concentrations, p47phox inhibition, and intracellular radical scavenging.

## Figures and Tables

**Figure 1 biomedicines-11-02877-f001:**
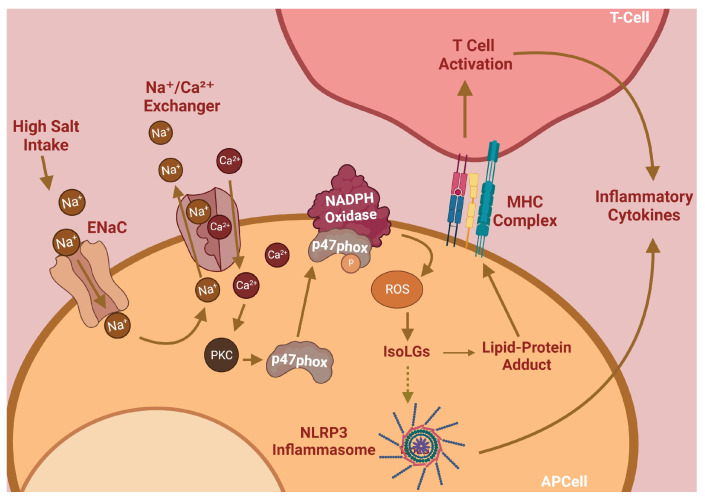
Salt-sensitive hypertension pathway.

**Figure 2 biomedicines-11-02877-f002:**
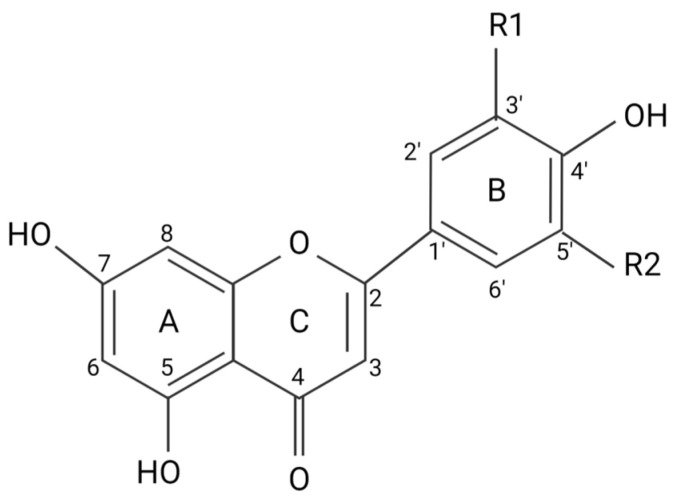
Flavonoid carbon numbers and ring names. (**A**) benzene ring; (**B**) phenyl ring; (**C**) heterocyclic ring.

**Figure 3 biomedicines-11-02877-f003:**
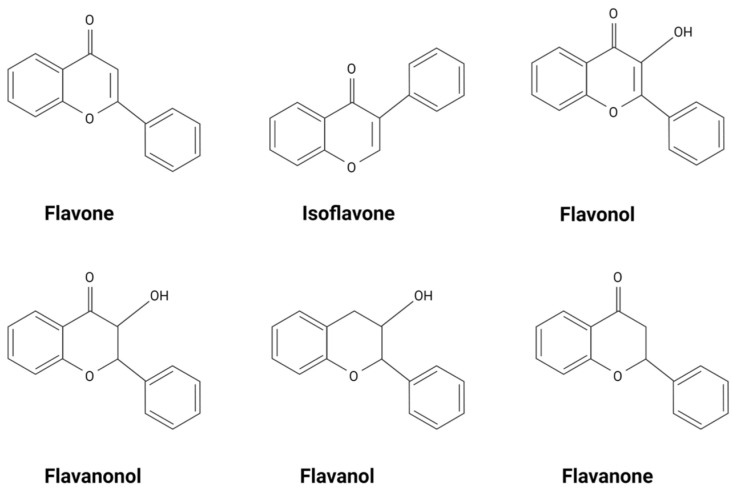
Types of flavonoids.

**Figure 4 biomedicines-11-02877-f004:**
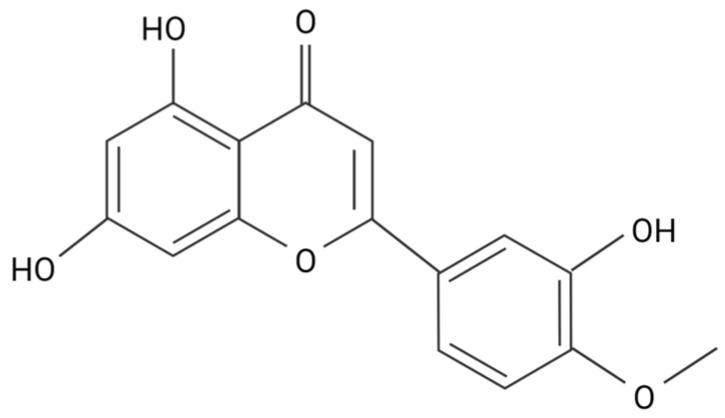
Structure of diosmetin.

**Figure 5 biomedicines-11-02877-f005:**
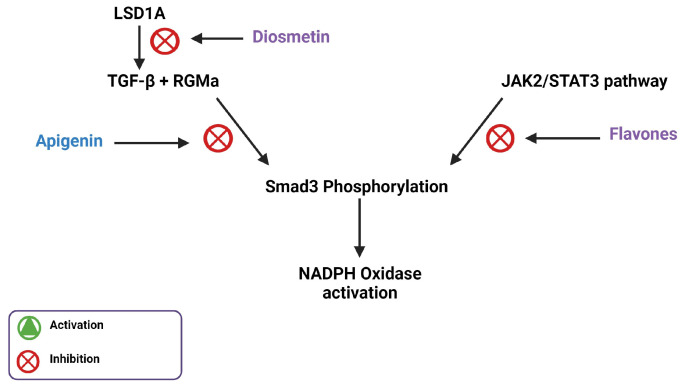
Illustration of Smad3 phosphorylation inhibition by flavones through the inhibition of LSD1A and the JAK2/STAT3 pathway.

**Figure 6 biomedicines-11-02877-f006:**
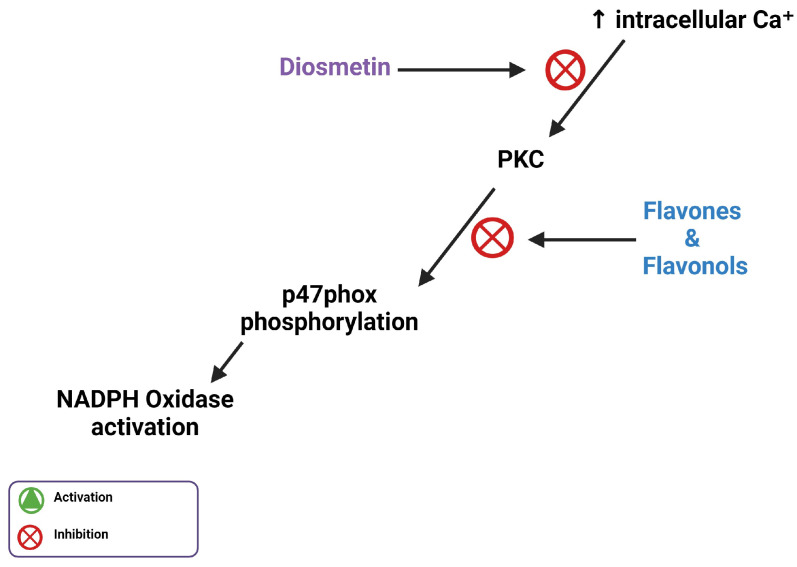
Illustration of flavones inhibiting increased intracellular calcium ion concentration through inhibition of PKC-mediated phosphorylation of p47phox.

**Figure 7 biomedicines-11-02877-f007:**
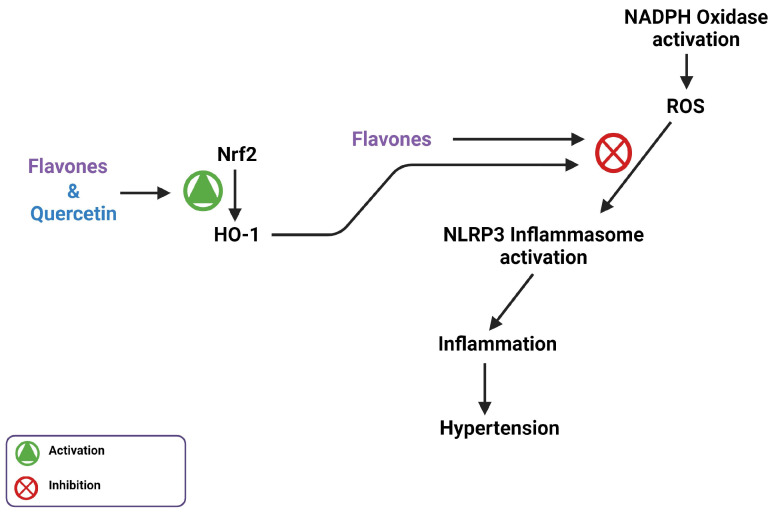
Flavonoids play a role in scavenging ROS through the activation of the Nrf2/HO-1 pathway and potentially direct intracellular ROS scavenging.

**Figure 8 biomedicines-11-02877-f008:**
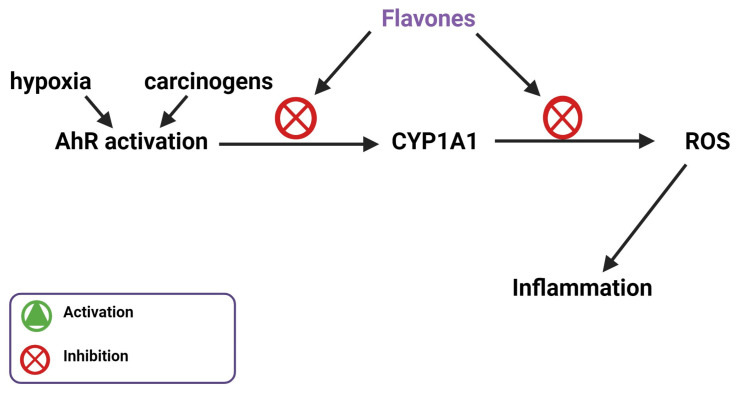
Flavones inhibit hypertension associated inflammation by inhibiting AhR and CYP1A1.

**Figure 9 biomedicines-11-02877-f009:**
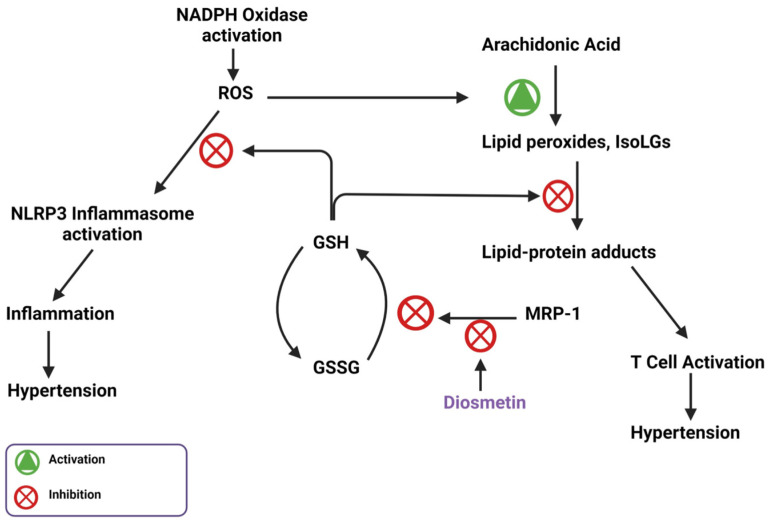
MRP-1 stops the intracellular anti-inflammatory response through inhibition of GSH-mediated ROS and lipid peroxide scavenging; inhibition of MRP-1 by diosmetin supports this intracellular anti-inflammatory response.

**Figure 10 biomedicines-11-02877-f010:**
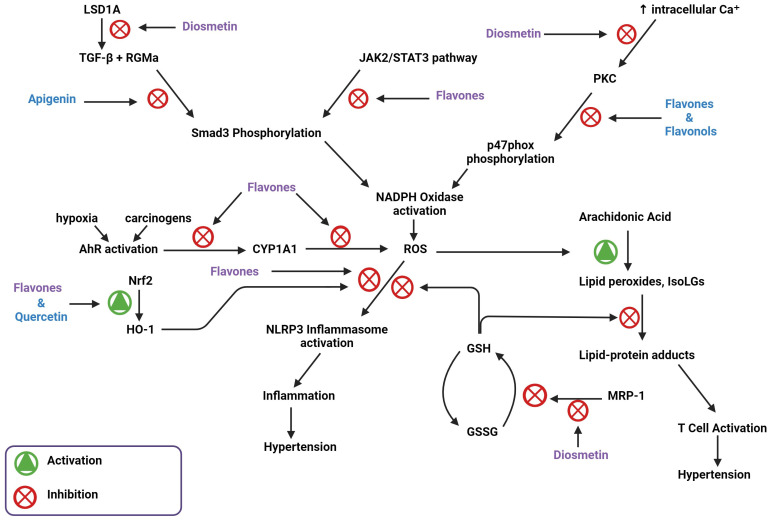
Flavone activity in different hypertensive pathways and potential hypertension-inhibiting mechanisms.

## Data Availability

No new data were created or analyzed in this study. Data sharing is not applicable to this article.

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
