# Peer review of "The Antioxidative Effects of Flavones in Hypertensive Disease"

_biomedicines, 2023, doi:10.3390/biomedicines11112877_

Round 1
Reviewer 1 Report
With the knowledge that hypertension is an important risk factor cardiovascular diseases, this paper “The anti-inflammatory effects of flavones in hypertensive disease”, reviewed the role of flavones, discussed the pharmacologic properties and potential therapeutic mechanisms against hypertension.
The topic is important and the manuscript provides a good analysis of the subject. I would recommend this manuscript after the following suggestions have been attended to:
The title mentions flavones and not diosmetin, thus, all chapters should replace diosmetin with flavones (e.g., “3. Flavones target...” instead of “3. Diosmetin targets...”; “5. Flavones inhibit...” instead of “5. Diosmetin inhibits...”, and so on)
Therefore, the chapters can be enhanced by discussing other flavones and mechanisms; following are a few examples that can be added:
- as flavonoids antiradical capacity is structure-dependent, C4' hydroxyl group of flavones and C3 hydroxyl group of flavonols can be associated with stronger scavenging potential (Spiegel et al. Flavones' and Flavonols' Antiradical Structure-Activity Relationship-A Quantum Chemical Study. Antioxidants (Basel). 2020 May 27;9(6):461. doi: 10.3390/antiox9060461).
- apigenin (4', 5, 7-trihydroxyflavone) improves hypertension and cardiac hypertrophy by modulating NADPH oxidase-dependent reactive oxygen species generation and inflammation (Gao et al. Apigenin Improves Hypertension and Cardiac Hypertrophy Through Modulating NADPH Oxidase-Dependent ROS Generation and Cytokines in Hypothalamic Paraventricular Nucleus. Cardiovasc Toxicol. 2021 Sep;21(9):721-736. doi: 10.1007/s12012-021-09662-1).
- genkwanin (4′,5-dihydroxy-7-methoxyflavone), a flavone with antioxidant and anti-inflammatory properties that inhibit ROS production and can present anticancer activity by causing cell cycle arrest, promoting cell apoptosis and autophagy, and inhibiting the expression of the PI3K/Akt signaling pathway (El Menyiy et al. Genkwanin: An emerging natural compound with multifaceted pharmacological effects. Biomed Pharmacother. 2023 Sep;165:115159. doi: 10.1016/j.biopha.2023.115159).
- possible side effects could arise from targeting NOX2, including the possibility that such inhibition can contribute to increased infections and/or autoimmune disorders (Nocella et al. Structure, Activation, and Regulation of NOX2: At the Crossroad between the Innate Immunity and Oxidative Stress-Mediated Pathologies. Antioxidants (Basel). 2023 Feb 9;12(2):429. doi: 10.3390/antiox12020429).
Author Response
With the knowledge that hypertension is an important risk factor cardiovascular diseases, this paper “The anti-inflammatory effects of flavones in hypertensive disease”, reviewed the role of flavones, discussed the pharmacologic properties and potential therapeutic mechanisms against hypertension.
The topic is important and the manuscript provides a good analysis of the subject. I would recommend this manuscript after the following suggestions have been attended to:
The title mentions flavones and not diosmetin, thus, all chapters should replace diosmetin with flavones (e.g., “3. Flavones target...” instead of “3. Diosmetin targets...”; “5. Flavones inhibit...” instead of “5. Diosmetin inhibits...”, and so on)
Therefore, the chapters can be enhanced by discussing other flavones and mechanisms; following are a few examples that can be added:
We thank the reviewer for taking the time to review this manuscript and appreciate their excellent insight. As suggested, we’ve adjusted each chapter to focus on flavones.
- as flavonoids antiradical capacity is structure-dependent, C4' hydroxyl group of flavones and C3 hydroxyl group of flavonols can be associated with stronger scavenging potential (Spiegel et al. Flavones' and Flavonols' Antiradical Structure-Activity Relationship-A Quantum Chemical Study. Antioxidants (Basel). 2020 May 27;9(6):461. doi: 10.3390/antiox9060461).
This is an excellent point by the reviewer. We’ve added this point in Lines 78-82, as well as used it to inform additional discussion added throughout the manuscript.
- apigenin (4', 5, 7-trihydroxyflavone) improves hypertension and cardiac hypertrophy by modulating NADPH oxidase-dependent reactive oxygen species generation and inflammation (Gao et al. Apigenin Improves Hypertension and Cardiac Hypertrophy Through Modulating NADPH Oxidase-Dependent ROS Generation and Cytokines in Hypothalamic Paraventricular Nucleus. Cardiovasc Toxicol. 2021 Sep;21(9):721-736. doi: 10.1007/s12012-021-09662-1).
This is an excellent resource and we’ve added it (Lines 149-153) to broaden this discussion around apigenin.
- genkwanin (4′,5-dihydroxy-7-methoxyflavone), a flavone with antioxidant and anti-inflammatory properties that inhibit ROS production and can present anticancer activity by causing cell cycle arrest, promoting cell apoptosis and autophagy, and inhibiting the expression of the PI3K/Akt signaling pathway (El Menyiy et al. Genkwanin: An emerging natural compound with multifaceted pharmacological effects. Biomed Pharmacother. 2023 Sep;165:115159. doi: 10.1016/j.biopha.2023.115159).
Again, this is an excellent point by the reviewer, and we’ve added discussion around inhibitory factors of ROS production associated with genkwanin (lines 238-244).
- possible side effects could arise from targeting NOX2, including the possibility that such inhibition can contribute to increased infections and/or autoimmune disorders (Nocella et al. Structure, Activation, and Regulation of NOX2: At the Crossroad between the Innate Immunity and Oxidative Stress-Mediated Pathologies. Antioxidants (Basel). 2023 Feb 9;12(2):429. doi: 10.3390/antiox12020429).
This is an excellent and relevant point the reviewer is correct in us not adequately addressing. We’ve added consideration of this side effect and other side effects discussed by Nocella et al. In Lines 380-398.
Reviewer 2 Report
The title of the paper is "The anti-inflammatory effects of flavones in hypertensive disease." However, the content discusses various oxidative stress-related pathways. It is suggested that the authors consider revising the title or placing a greater emphasis on anti-inflammatory pathways.
It is recommended to remove molecular structure diagrams that are unrelated to flavones' components, such as Figure 5 and Figure 7, as they have less direct relevance to the mechanisms of flavones.
Please to be mindful of the point that you are trying to make. It appears that there is very little critical appraisal throughout the document.
Author Response
The title of the paper is "The anti-inflammatory effects of flavones in hypertensive disease." However, the content discusses various oxidative stress-related pathways. It is suggested that the authors consider revising the title or placing a greater emphasis on anti-inflammatory pathways.
We thank the reviewer for this thoughtful comment. Based on this comment, we have changed the title to antioxidative to better reflect the content of the review.
It is recommended to remove molecular structure diagrams that are unrelated to flavones' components, such as Figure 5 and Figure 7, as they have less direct relevance to the mechanisms of flavones.
This is an excellent point by the reviewer. In response, we have modified these figures to make them more relevant to the overall point of the manuscript.
Please to be mindful of the point that you are trying to make. It appears that there is very little critical appraisal throughout the document.
We thank the reviewer for this comment and the time taken to offer this insight. Throughout the manuscript, we’ve focused on offering greater critical appraisal (see text highlighted yellow).
Round 2
Reviewer 2 Report
The authors have addressed some of my comments, I think the authors have done a good job.
Author Response
The authors are thankful for Reviewer 2's constructive criticisms which we believe have significantly improved the manuscript's quality.